# Rhizosphere microbial community structure in high-producing, low-input switchgrass families

**Christina Stonoha-Arther** *◉, **Kevin Panke-Buisse**◉, **Alison J. Duff**◉,
**Andrew Molodchenko**◉, **Michael D. Casler**¤

USDA-ARS US Dairy Forage Research Center, Madison, WI, United States of America

◉ These authors contributed equally to this work.
¤ Current address: Department of Plant and Agroecosystem Sciences, University of Wisconsin, Madison, WI, United States of America
* christina.arther@usda.gov

**Data Availability Statement:** We have added our R scripts and sequencing data to a Zenodo repository (https://zenodo.org/doi/10.5281/zenodo.11169988).

## Abstract

Switchgrass (*Panicum virgatum* L.) is a native, low-input North American perennial crop primarily grown for bioenergy, livestock forage, and industrial fiber. To achieve no-input switchgrass production that meets biomass needs, several switchgrass genotypes have been identified that have a low or negative response to nitrogen fertilizer, *i.e.*, the biomass accumulation with added nitrogen is less than or equal to that when grown without nitrogen. In order to improve the viability of low-input switchgrass production, a more detailed understanding of the biogeochemical mechanisms active in these select genotypes is needed. 16S and ITS amplicon sequencing and qPCR of key functional genes were applied to switchgrass rhizospheres to elucidate microbial community structure in high-producing, no-input switchgrass families. Rhizosphere microbial community structure differed strongly between sites, and nitrogen responsiveness.

## Introduction

Switchgrass (*Panicum virgatum* L.) is a native, low-input North American perennial forage crop, and is also of interest as a lignocellulosic bioenergy feedstock [1]. Decreasing both the environmental impact and the cost of switchgrass production continues to be a priority to maximize crop utilization and profitability. Decreasing nitrogen inputs has been a goal of breeders and agronomists. Several studies and meta-analyses have identified switchgrass cultivars that do not respond to nitrogen fertilization. These results indicate that breeding for decreased nitrogen input requirements is possible and raises questions about the mechanisms of nitrogen acquisition in these genotypes; specifically, how soil microbes influence the nitrogen status of switchgrass [2–4].

Switchgrass cultivars respond to nitrogen fertilizer application to varying degrees, with some cultivars not responding at all; furthermore, this effect was location dependent [2–5]. This lack of response to added nitrogen in some cultivars has sparked an interest in studying the microbial community associated with the roots and rhizosphere to identify microbes that

**Funding:** This research was based upon work supported in part by the Great Lakes Bioenergy Research Center, U.S. Department of Energy, Office of Science, Office of Biological and Environmental Research under Award Numbers DE-SC0018409. The research was conducted while the author was an employee of USDA-ARS and partly funded by congressionally allocated funds to the U.S. Dairy Forage Research Center. The funders had no role in study design, data collection and analysis, decision to publish, or preparation of the manuscript.

**Competing interests:** The authors have declared that no competing interests exist.

may explain this phenomenon. Associative nitrogen fixation has been identified as a source of nitrogen in switchgrass and was found to be somewhat cultivar specific [6–8]. Furthermore, arbuscular mycorrhizal fungi (AMF) species richness in switchgrass was found to be influenced by nitrogen fertilization regimen, cultivar, and environmental conditions [9]. The switchgrass microbiome was found to be strongly associated with location, but it was also influenced by genotype [10]. Taken together, these studies suggest that the microbial community plays an important role in switchgrass nitrogen acquisition in a genotype- and location-specific manner. Further interrogation of the switchgrass microbiome may reveal the members and function of the microbial community that influences the nitrogen status of the non-responding genotypes.

Microbes play an important role in the nitrogen status of plants, especially in the absence of inorganic nitrogen fertilizer. Different microbes can affect nitrogen availability in different ways [11]. For example, bacterial diazotrophs fix atmospheric dinitrogen gas into ammonium and can either be symbiotic (as is the case for many legumes), associative, or free-living. There are many examples of associative nitrogen fixation in grasses, which can enrich the rhizosphere with the appropriate diazotrophs [12]. Often, these associative diazotrophs are specific strains that carry the nitrogenase gene (*nifH*); nitrogen fixation genes can be horizontally transferred between bacteria, or lost within lineages [13]. Therefore, the presence and quantity of *nifH* copies in the rhizosphere can be used as a proxy for the ability of the bacterial population to fix nitrogen [14].

Arbuscular mycorrhizal fungi (AMF) can also play an important role in the nitrogen status of plants. AMF can supply the plant with soil nitrogen that would otherwise be unavailable, and higher AM colonization can increase the nitrogen that is transferred from the soil to the plant [15]. AMF colonize the roots of many crops, including grasses, and increase the surface area of the roots to reach nitrogen that the roots cannot [11, 16]. These fungi can transport both inorganic and organic nitrogen from the soil to the plant. For example, two ammonium transporters (named AMT1 and AMT2) have been characterized in *Rhizophagus irregularis*, a common AMF [16, 17]. Presence of *AMT1* in the rhizosphere could help predict the AMF colonization level, as well as the capacity of these fungi to transport nitrogen from the soil.

Non-symbiotic microbes can also influence the availability of nitrogen in the soil via their role in biogeochemical cycling. For example, saprotrophic fungi and bacteria decompose organic matter, which releases nitrogen that can then be taken up by plant roots or AMF, which are not saprotrophic. However, negative effects of the microbial community cannot be overlooked. Temporal, negative priming of soil nutrients can occur upon the addition of various carbon or nitrogen sources due to the actions of soil microorganisms [18]. When interrogating the microbial community for the ability to influence the nitrogen status of plants, it is important to consider both the positive and negative effects of symbiotic and non-symbiotic bacteria and fungi.

One reason that switchgrass is an attractive bioenergy crop is because it produces biomass on marginal soils. In order to further improve switchgrass for this purpose, it has been suggested that switchgrass should be bred to require less nitrogen while concomitantly selected for increased biomass in order to decrease the nitrogen removal that will inevitably increase if it is bred for biomass only [19]. In the present study, the rhizosphere microbial community was characterized in a subset of switchgrass families that were previously found to have no or minimal response to nitrogen fertilizer [4].

The objectives of this study were the following: to characterize the switchgrass rhizosphere microbiome with and without added nitrogen fertilizer; to identify microbial candidates that may contribute to the biomass accumulation of switchgrass genotypes that do not respond to nitrogen fertilizer; and to identify any microbes that may be preventing the non-responsive

switchgrass genotypes from utilizing nitrogen fertilizer. We hypothesized that specific microbes recruited by the non-responding switchgrass families may help provide these genotypes with nitrogen, providing a productivity benefit in the absence of nitrogen fertilizer.

## Materials and methods

### Site description

Switchgrass plots were established in May 2018 at two sites in Wisconsin, USA to evaluate biomass production with and without nitrogen fertilizer [4]. The paired-plot augmented split-block design included 12 blocks per location and 20 whole plots per block. A total of 83 half-sib families were evaluated in the nitrogen experiment. Within each block, two families with the highest yield under no nitrogen compared to yield under nitrogen (in the form of ammonium nitrate) applied at 100 kg/ha (hereafter referred to as N 100) was identified within each block. The paired plots (N 0, N 100) associated with these high-performing families were selected for soil rhizosphere sampling. Families were categorized as responsive or non-responsive to nitrogen, according to whether their biomass yield under N 0 was less than or greater than their biomass yield under N 100, respectively (S1 Table; [4]).

### Soil sampling and chemical properties

Soil samples were collected for biological analyses in July 2022. At both sites, all sampling and processing tools were sprayed with 70% ethanol and allowed to dry prior to each soil sample collection. Field personnel wore clean nitrile gloves for processing each sample. Soil samples were immediately stored in a cooler in the field and transferred to a -80°C freezer the same day.

The Hancock site (44.12°N, 89.54°W) soil type is a Plainfield loamy sand (mixed, mesic Typic Udipsamment). A shovel was used to collect a root ball to 30 cm depth. Loose soil was shaken and kneaded from the root ball, and rhizosphere soil was then separated from the roots and transferred to a sample bag.

The Prairie du Sac site (43.45°N, 89.76°W) soil type is a Richwood silt loam (fine-silty, mixed, superactive, mesic Typic Argiudoll). Due to difficulty hand sampling in this soil type in dry conditions, samples were extracted with a steel probe mounted on a hydraulic Giddings sampler (Windsor, CO). A 7.62 cm diameter soil core was extracted from 30 cm depth, and loose soil was shaken and kneaded from the roots. Rhizosphere soil was separated from the roots and then transferred to a sample bag. The samples were ultimately placed in a -80°C freezer for long-term storage and stored at -20°C when in use.

Total carbon and total nitrogen were measured in rhizosphere soil samples via combustion analysis on a LECO TruMac CN instrument (St. Joseph, Michigan). Soil moisture, total carbon, and total nitrogen are reported in Table 1. Field pH values were reported previously and were measured throughout the course of the study from 2018 to 2022 [4].

**Table 1. Soil characteristics by location.**

| Site | Moisture | Total Nitrogen | Total Carbon |
|------|----------|----------------|--------------|
| | (%) | (mg/g) | (mg/g) |
| PDS | 15.49 ± 1.49 | 0.68 ± 0.44 | 7.05 ± 1.82 |
| HAN | 5.97 ± 0.37 | 0.32 ± 0.20 | 3.56 ± 0.28 |

PDS, Prairie du Sac; HAN, Hancock; Values presented as mean ± standard deviation; n = 48. All measurements were statistically significant between the two sites p < .00001.

## Real-time PCR of *nifH* and *AMT1*

For real-time quantitative PCR (qPCR), DNA was extracted from the collected soil samples using the Zymo Quick-DNA Fecal/Soil Microbe 96 Kit (Irvine, CA; catalog no. D6011) following the manufacturer's protocol. Zymo Quick-DNA Fecal/Soil Microbe Miniprep Kit (Irvine, CA; catalog no. D6010) was used on any samples that needed to be re-extracted. *Ensifer meliloti* rhizobia strain Rm1021 was grown for two days on tryptone-yeast media at 30˚C. *E. meliloti* DNA was extracted using the NucleoSpin Tissue Kit from Machery-Nagel (Düren, Germany; catalog no. 740952.50), according to the manufacturer's instructions. DNA concentration of the DNA samples was determined using the high sensitivity Quant-iT dsDNA Assay Kit from Thermo Fisher (Waltham, MA; catalog no. Q33120) on a Promega GloMax plate reader (Madison, WI).

Quantitative PCR was performed on the collected soil samples, using the *E. meliloti* DNA as a known standard. qPCR was done on the bacterial *nifH* gene using the degenerate primer pair IGK3/DVV [20, 21]. The qPCR reactions were 20 $\mu$L including 500 nM concentration of each primer, 2 $\mu$L of template DNA, and 6 $\mu$L of Thermo Fisher PowerUp SYBR Green Master Mix (Waltham, MA). The PCR was carried out on an Applied Biosystems QuantStudio 5 Real-Time PCR machine (Waltham, MA).

The qPCR data were analyzed using the One-Point Calibration (OPC) method as previously described, using the *E. meliloti* DNA as the calibration point [22]. As per the OPC method, the amplification efficiency was estimated from the amplification plot using the web based LinRegPCR program [23]. qPCR of each sample was done in triplicate or quadruplicate; the data were included in the analysis if the standard deviation of the Cq values of the technical replicates was less than or equal to 0.25.

Degenerate primers of *AMT1* in AMF were designed with the following: *AMT1* was accessed through Genbank (accession AJ880327.1) and then a BLAST search was performed on MYCOCOSM [24–26]. *AMT1* genes from the following AMF species accessions were used to design the primers: *Rhizophagus intraradices* DAOM197198, *Rhizophagus clarus* HR1, *Rhizophagus irregularis* DAOM 181602 v1.0. The AMT1 sequences were aligned with MUSCLE and then the locations of the primers were chosen with Primer-BLAST [27, 28]. Only primers that were present in alignments of all 3 species were chosen. We used degeneracy codes and manually designed primers with degeneracies in positions varying between sequences (S2 Table).

Quantitative PCR was performed on the collected soil samples, using the *R. irregularis* DNA as a known standard. The *R. irregularis* DNA was extracted with Quick-DNA Fecal/Soil Microbe Miniprep Kit (Zymo Research, CA) from aseptic spores PTB297-L-ASP-A, 4000 spores/mL (Premier Tech Biotechnologies, Quebeck).

The qPCR reactions were 20 $\mu$L including 500 nM concentration of each primer, 2 $\mu$L of template DNA, and 6 $\mu$L of Thermo Fisher PowerUp SYBR Green Master Mix (Waltham, MA). The PCR was carried out on an Applied Biosystems QuantStudio 5 Real-Time PCR machine (Waltham, MA), with an annealing temperature of 56˚C.

The qPCR data were analyzed using the One-Point Calibration (OPC) method as described by Brankatschk and colleagues (2012) [22], using *R. irregularis* DNA as the calibration point. As per the OPC method, the amplification efficiency was estimated from the amplification plot using the web based LinRegPCR program [23]. qPCR of each sample was done in triplicate; the data were included in the analysis if the standard deviation of the Cq values of the technical replicates was less than or equal to 0.30. We could not achieve the mentioned standard deviation of Cq values for some samples; in that case, we combined data from several runs with SD less than 0.90.

## Metagenomic sequencing and analysis

For microbial sequencing, DNA was extracted from the collected soil samples using the Quick-DNA Fecal/Soil Microbe 96 Kit (Irvine, CA; catalog no. D6011) following the manufacturer's protocol. Purified genomic DNA was submitted to the University of Wisconsin-Madison Biotechnology Center. DNA concentration was verified fluorometrically using either the Qubit® dsDNA HS Assay Kit or Quant-iT™ PicoGreen® dsDNA Assay Kit (Thermo-Fisher Scientific, Waltham, MA, USA). Samples were prepared in a similar process to the one described in Illumina's 16s Metagenomic Sequencing Library Preparation Protocol, Part # 15044223 Rev. B (Illumina Inc., San Diego, California, USA) with the following modifications: The ITS region was amplified with fusion primers (S2 Table). Region specific primers were previously described (ITS1-F [29]; ITS4 [30]) and were modified to add Illumina adapter overhang nucleotide sequences to the region-specific sequences. Following initial amplification, reactions were cleaned using a 0.7x volume of AxyPrep Mag PCR clean-up beads (Axygen Biosciences, Union City, CA). Using the initial amplification products as template, a second PCR was performed with primers that contain Illumina dual indexes and Sequencing adapters (S2 Table, where bracketed sequences are equivalent to the Illumina Dual Index adapters D501-D508 and D701-D712, N716, N718-N724, N726-N729). Following PCR, reactions were cleaned using a 0.7x volume of AxyPrep Mag PCR clean-up beads (Axygen Biosciences). Quality and quantity of the finished libraries were assessed using an Agilent DNA 1000 kit (Agilent Technologies, Santa Clara, CA) and Qubit® dsDNA HS Assay Kit (ThermoFisher Scientific), respectively. Libraries were pooled in an equimolar fashion and appropriately diluted prior to sequencing. Paired end, 300 bp sequencing was performed using the Illumina MiSeq Sequencer and a MiSeq 600 bp (v3) sequencing cartridge.

Amplicon sequence analysis and data visualizations were performed in R. 16S and ITS amplicon sequence joining, denoising, quality filtration, and taxonomic classification were performed according to default parameters of the dada2 package [31]. The phyloseq and ggplot2 packages were used for alpha diversity calculation and visualization [32, 33]. R scripts and data used in this study are available at: https://zenodo.org/doi/10.5281/zenodo.11169988. Phylogenetic trees were built with the FastTree 2 program [34]. Differential abundance analyses were performed with the DESeq2 package [35]. Venn diagrams were created with the Russell88/MicEco package (https://zenodo.org/record/4733747).

## Results

### Switchgrass rhizosphere carbon, nitrogen, and species richness

Switchgrass families grown in two different locations in Wisconsin, Prairie du Sac (PDS) and Hancock (HAN), were subjected to two different nitrogen fertilizer regimens (N 0, N 100). The yield of these families was measured and labeled as either nitrogen non-responsive (no yield increase when grown with nitrogen fertilizer), or nitrogen responsive (yield increase when grown with nitrogen; [4]). Soil moisture, total carbon, and total nitrogen were significantly different between the two locations, HAN and PDS, but no other differences were noted (Table 1).

16S and ITS sequencing revealed trends in bacterial and fungal alpha diversity across switchgrass families by Inverse Simpson and Fisher's Alpha indices (Fig 1). Overall, the PDS location had higher bacterial alpha diversity compared to the HAN location, according to the Fisher diversity metric (p = 0.000246). Notably, fungal species richness differed between responders treated with 100 N and non-responders in the 0 N treatment at both PDS and HAN (p = 0.00143, Fisher diversity metric; Fig 1).

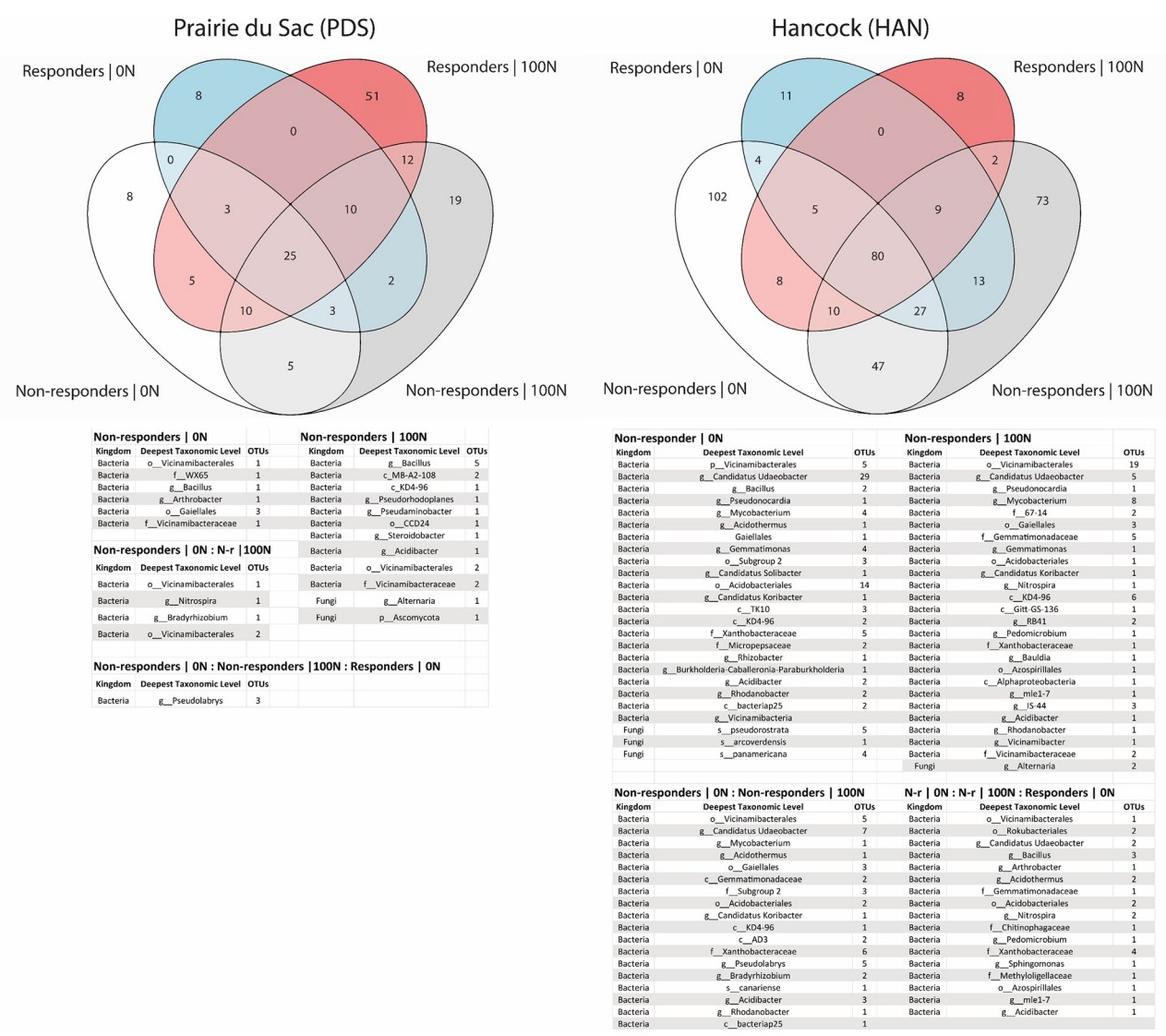

**Fig 1. Species richness.** Comparison of species richness for both sites (left and right columns), bacteria (top row), and fungi (bottom row) as determined by amplicon sequencing of 16S and ITS, respectively. The right-hand legend is common for all plots. Mean comparison between groups was via ANOVA with post-hoc Tukey's HSD. There were no significant differences between groups in bacterial richness despite minor numerical trends (PDS p = 0.367, HAN p = 0.211). Fungal community richness differed significantly between fertilized responders at PDS and Hancock and unfertilized non-responders for the Fisher diversity metric (p = 0.00143), but not for the Inverse Simpson metric (p = 0.181).

## Rhizosphere community composition

Unique bacterial and fungal taxa within and between the families at each location were depicted with Venn diagrams (Fig 2). There were more unique taxa seen in every relevant group at the HAN location compared to the PDS location. Patterns of taxa membership across groups were used to interrogate potential taxa of interest related to nitrogen response dynamics in switchgrass families.

## *nifH* gene abundance

*nifH* gene abundance was used to predict the ability of the rhizosphere soil community to fix nitrogen for the plant. We were able to quantify *nifH* copy number from every soil sample

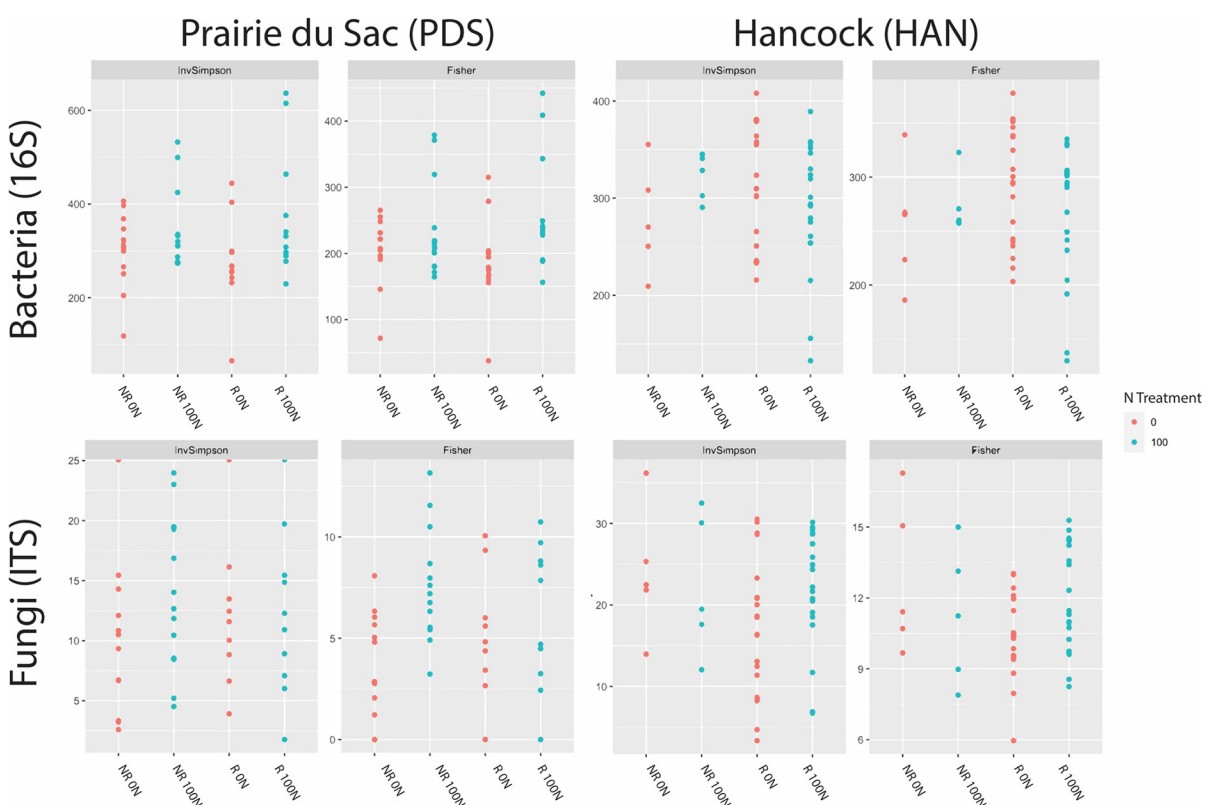

**Fig 2. Community assembly Venn diagram.** Comparison of amplicon sequence variants (ASVs) between groups and sites. ASVs are unique sequences identified in the dataset and can be compared to operational taxonomic units (OTUs) with no clustering. Only ASVs present in 80 percent of samples for a given group were included. Tables below each Venn diagram contain the deepest level of assigned taxonomy and the number of ASVs assigned to that taxon for relevant groups and overlaps.

except three, which had too much variation in the technical replicates to be reliable. We found no significant differences in the *nifH* abundance between the switchgrass families grown with or without nitrogen fertilizer, or between the two locations. Furthermore, we did not observe a trend correlating switchgrass response to nitrogen fertilizer and *nifH* gene abundance, providing no evidence of a difference in diazotroph rhizosphere abundance between groups (S1 File).

## Arbuscular mycorrhizal *AMT1* gene abundance

In order to include a proxy for the abundance of AMF species that have the ability to transport ammonium from the soil, we designed degenerate primers for the *AMT1* gene that has been implicated in ammonium transport from the soil to the fungus [17]. We chose rhizosphere samples from five switchgrass families grown at the PDS location and four families grown at the HAN location, each grown under the 0 N condition and the 100 N condition. There was no correlation between the *AMT1* copy number per nanogram of DNA and the family response to nitrogen, but there was a possible trend at HAN of increased *AMT1* copy number in the 100 N treated rhizosphere compared to the 0 N rhizosphere (p = .087, Student's t-test; S1 Fig).

## Discussion

This study found that the rhizosphere microbial community and soil chemical properties differed strongly between the two Wisconsin locations, PDS and HAN. This agrees with many

other studies, and also highlights the findings that location is the biggest driver of microbial community assembly [10]. The difference in microbial communities and soil chemical properties between the two locations may also help explain why different families had different responses to nitrogen at PDS and HAN [4].

Patterns of taxa presence and absence across treatment and response groups were used to identify taxa associated with observed trends in nitrogen responsiveness across switchgrass families. Specifically, taxa belonging exclusively to three membership groups of interest were identified: 1. 0 N non-responding rhizospheres; 2. 0 N and 100 N non-responding rhizospheres; and 3. 0 N and 100 N non-responding rhizospheres and 0 N responding rhizospheres. Specific microbes recruited by the non-responding switchgrass families may help provide these genotypes with nitrogen, providing a fitness benefit in the absence of nitrogen fertilizer. Taxa exclusively occurring within the 0 N non-responsive group may indicate taxa that potentially contribute to the ability of these specific switchgrass families to efficiently accumulate biomass in the absence of added nitrogen. This group consisted of eight unique taxa (bacterial and fungal) at PDS and 102 at HAN. Taxa unique to the rhizosphere of nitrogen non-responders in both the 0 N and 100 N treatments were of interest because these families may be recruiting microbes that help them acquire nitrogen in a genotype-specific way, independent of the nitrogen treatment. There were no fungal taxa that fit this criterion in the switchgrass grown at either location. At PDS, there were five unique bacterial taxa within this group, compared to 47 taxa from the families grown at HAN. At both locations, there were taxa that belonged to the *Bradyrhizobium* genus found within this category, albeit they were identified as different operational taxonomic units (OTUs) between the two locations. Of note, there was also a member of the genus *Nitrospira* within this category found in the families grown at the PDS location.

The last category that was examined in detail was the unique taxa found between three groups: non-responders under both nitrogen treatments, and the responders with 0 N nitrogen treatment. In other words, all of the groups except the responding families that were treated with 100 N. Taxa associated with non-responsiveness in both treatments and 0 N responders that are present in all genotypes and may help with nitrogen acquisition but disappear upon fertilization of the nitrogen responders. Alternatively, the presence of these taxa in the non-responding 100 N group could hinder their ability to effectively utilize the added nitrogen fertilizer. At PDS, there were three bacterial taxa that fell into this group, compared to 27 at HAN. The same *Nitrospira* OTU mentioned above was also found within this group at the HAN location.

At both locations *Bradyrhizobium spp.* were unique among the non-responders (two OTUs at HAN and one at PDS). *Bradyrhizobium* species primarily fix nitrogen in close relationship with legumes, but there are some examples of members of this genus being free-living diazotrophs [36, 37]. Association with free-living diazotrophs could potentially provide a benefit to the plant host in the absence of added nitrogen. However, the *nifH* gene abundance data do not support a significant enrichment of *nifH*-carrying bacteria (S1 File). It is therefore possible that these *Bradyrhizobium* strains are indeed fixing nitrogen for the plants, but their presence does not significantly enhance the *nifH* gene abundance in the rhizosphere. However, presence of DNA does not necessarily indicate a difference in expression of the gene. *nifH* gene expression could be higher in the non-responders and receive greater amounts of fixed nitrogen despite the lack of difference in *nifH* copy number or gene abundance. These *Bradyrhizobium spp.* may also be providing another benefit to the plants aside from nitrogen fixation. Further functional studies need to be carried out to determine which of these hypotheses are relevant.

One interesting taxon that was absent in the N 100 soil responder group, but present in the non-responders at both locations is an OTU belonging to the genus *Nitrospira*. *Nitrospira*

species are mostly nitrite-oxidizing bacteria that play a pivotal role in nitrification in the soil [38]. It is estimated that actions of nitrifying bacteria in the soil can result in a 50% loss of plant-usable nitrogen [39]. The observed *Nitrospira* taxon is conspicuously absent in the rhizosphere of the families that benefit from nitrogen fertilization, which may be indicative of nitrification playing a role in response patterns.

Saprotrophs have also been shown to increase the availability of nitrogen to plants, by freeing nitrogen from organic matter [11, 40]. It is therefore possible that some of the taxa specific to the non-responders are performing this function. This would be somewhat difficult to predict because many saprotrophs exist in the soil. For example, at HAN, nine unique fungal taxa were found in the 0 N non-responder group. Four of these OTUs belonged to the genus *Baudoinia*, which includes saprotrophs that have been found to use ethanol, glucose, and acetate as carbon sources in urban settings, but have generally been poorly characterized in soils [41]. Different saprotrophs could theoretically fill this role, depending on which microbes are present in the bulk soil.

It is also important to acknowledge that the mechanism driving some of the switchgrass families to be nitrogen non-responders could be a factor that prevents these families from effectively utilizing the added nitrogen. One possibility is the negative priming effect that occurs after a source of carbon or nitrogen is added to the soil. In the case of adding nitrogen to the soil, there have been several examples of N-immobilization by microorganisms, depending on the soil type and carbon content [18]. It is therefore possible that the added nitrogen is being captured by the microbes in the soil, causing it to be unavailable to the plants.

One unexpected finding in this study is the apparent increase in the AMF *AMT1* gene in the rhizosphere of 100 N treated soil at the HAN location. It was hypothesized that AMF richness and abundance would decrease upon nitrogen fertilization, although one study found that AMF colonization was the same when nitrogen was added to the soil [9]. However, other studies have demonstrated that AMF can function as commensalists or parasites when plants are grown under nitrogen limiting conditions and the effect was dependent on plant, and, presumably, fungi genotypes [42, 43]. At the HAN location, the interactions between AMF and these switchgrass genotypes may have been negative under low nitrogen conditions and positive under higher nitrogen conditions, thus incentivizing the plants to limit fungal colonization in the 0 N condition, but not the 100 N condition. Future studies exploring these findings would benefit from directly measuring AMF fungal colonization in the plants.

The most likely explanation for the observation that some switchgrass families do not respond to nitrogen fertilization is a complicated interaction between location, plant genotype, and soil microbial community. This is highlighted by the large differences in microbial community between locations, and that different families perform differently at each location. Future studies will benefit from measuring functional traits of the microbial community in addition to taxonomic estimation. The present study is limited by a need for deeper understanding of what services the rhizosphere microbiome may be performing for the plant. A more thorough investigation may include measuring nitrogen fixation, AMF colonization in the roots, and nitrogen mineralization in the soil. Along with this study, future research may be able to predict which switchgrass genotypes will perform best at a specific location with the lowest amount of nitrogen inputs.

This study may help generate ideas on how to breed generalist switchgrass cultivars to thrive without any nitrogen fertilizer, which was proposed in the paper by Casler (2023) [4]. The dilemma is that no one cultivar will be able to accumulate adequate biomass without nitrogen fertilizer; this ability is highly dependent on location and soil. The previous solution that was proposed was that breeders would use parental genotypes that were tested and selected at two or more locations. The progeny would then form a population that, as a whole,

would act as a generalist [4]. If soil microbes identified in this study are shown to provide a benefit to some of these switchgrass cultivars grown without nitrogen, perhaps a mixed-strain inoculum could be coated on the seeds and used for selecting the parent genotypes, as well as for the populations that will be planted. For example, if the multiple *Bradyrhizobium* strains identified in this study are shown to be beneficial, a mixture of these strains could be used to 1. select parental cultivars tested in multiple locations that may be able to use and benefit from the inoculated strains, and 2. as a seed inoculum to increase the favorable response of the populations of progeny ultimately used for biomass production.

## Supporting information

**S1 Fig. Arbuscular mycorrhiza *AMT1* gene quantification of different switchgrass families grown at two different locations in Wisconsin.** (A) Copy number of *AMT1* per nanogram of rhizosphere DNA of five different switchgrass families grown under 0 N and 100 N in Prairie du Sac, Wisconsin ($p > 0.1$, Student's t-test). (B) Copy number of *AMT1* per nanogram of rhizosphere DNA of four different switchgrass families grown under 0 N and 100 N in Hancock, Wisconsin ($p = 0.087$, Student's t-test). Error bars are standard error of the calculated copy number of *AMT1*. The families on the x-axis are arranged from least responsive to nitrogen to most responsive calculated from percent yield difference. Dark gray bars are families grown with 0 nitrogen; light gray bars are the same families grown with 100 kg/ha nitrogen.
(TIF)

**S1 Table. t-tests of biomass yield difference between switchgrass families chosen for this study treated with 100 kg/ha nitrogen and 0 kg/ha nitrogen at two different locations in Wisconsin (Prairie du Sac and Hancock).** Plants were chosen for sampling and analysis from Casler, 2023.
(PDF)

**S2 Table. Primers used in this study.**
(PDF)

**S1 File. *nifH* copy number per sample.**
(PDF)

## Author Contributions

**Conceptualization:** Kevin Panke-Buisse, Michael D. Casler.

**Data curation:** Christina Stonoha-Arther, Alison J. Duff.

**Formal analysis:** Christina Stonoha-Arther, Kevin Panke-Buisse, Andrew Molodchenko.

**Funding acquisition:** Michael D. Casler.

**Investigation:** Christina Stonoha-Arther, Kevin Panke-Buisse, Andrew Molodchenko.

**Methodology:** Christina Stonoha-Arther, Kevin Panke-Buisse, Andrew Molodchenko, Michael D. Casler.

**Project administration:** Alison J. Duff, Michael D. Casler.

**Resources:** Alison J. Duff, Michael D. Casler.

**Supervision:** Alison J. Duff.

**Visualization:** Kevin Panke-Buisse.

**Writing – original draft:** Christina Stonoha-Arther, Kevin Panke-Buisse.

**Writing – review & editing:** Christina Stonoha-Arther, Kevin Panke-Buisse, Alison J. Duff, Andrew Molodchenko, Michael D. Casler.

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
