## [Decision Letter · Decision Letter 0]

19 Apr 2024

PONE-D-24-09855Rhizosphere microbial community structure in high-producing, low-input switchgrass familiesPLOS ONE

Dear Dr. Stonoha-Arther,

Thank you for submitting your manuscript to PLOS ONE. After careful consideration, we feel that it has merit but does not fully meet PLOS ONE’s publication criteria as it currently stands. Therefore, we invite you to submit a revised version of the manuscript that addresses the points raised during the review process.

We look forward to receiving your revised manuscript.

Kind regards,

Mayank Gururani

Academic Editor

PLOS ONE

Journal Requirements:

   "This research was based upon work supported in part by the Great Lakes Bioenergy Research Center, U.S. Department of Energy, Office of Science, Office of Biological and Environmental Research under Award Numbers DE-SC0018409. The research was conducted while the author was an employee of USDA-ARS and partly funded by congressionally allocated funds to the U.S. Dairy Forage Research Center."

Reviewers' comments:

Reviewer's Responses to Questions

**Comments to the Author**

1. Is the manuscript technically sound, and do the data support the conclusions?

Reviewer #1: Yes

Reviewer #2: Partly

2. Has the statistical analysis been performed appropriately and rigorously? 

Reviewer #1: Yes

Reviewer #2: Yes

3. Have the authors made all data underlying the findings in their manuscript fully available?

Reviewer #1: Yes

Reviewer #2: Yes

4. Is the manuscript presented in an intelligible fashion and written in standard English?

Reviewer #1: Yes

Reviewer #2: Yes

5. Review Comments to the Author

Reviewer #1: Dear authors,

this is an interesting work and can be published in this journal.

please present your Rescripts for diversity analyses (phyloseq,..) in supplementary file (GitHub link) and refer in the materials and methods.

Best regard

Reviewer #2: The MS by Stonoha-Arther et. al investigates microbial communities of switchgrass rhizosphere in response to nitrogen availibility in two different locations.The study identifies unique microbial taxa associated with nitrogen non-responsive switchgrass families and highlights the complex interactions between plant genotype, location, and soil microbial communities in switchgrass nitrogen response, the leads from which can be adapted for future breeding programs to reduce nitrogen use in switchgrass cultivation. However, the MS in its present forms needs to be revised suitably before accepted for publication. The major and minor comments attached as a word document

6. PLOS authors have the option to publish the peer review history of their article (what does this mean?). If published, this will include your full peer review and any attached files.

Reviewer #1: No

Reviewer #2: No

---

## [Author Response · Author response to Decision Letter 0]

14 May 2024

Mayank Gururani and reviewers:

Thank you for your time and effort to improve this manuscript titled, “Rhizosphere microbial community structure in high-producing, low-input switchgrass families.” We sincerely appreciate the opportunity to edit this manuscript and we hope that we have addressed the comments to the satisfaction of the editors and reviewers. Here, we have included a point-by-point response to all of the comments that we have received. Changes are outlined here and in our marked-up copy of the manuscript. 

Editor comments:

Author response: Thank you for highlighting this. We have checked that our manuscript meets PLOS ONE’s style requirement. 

Author response: Thank you for pointing this out. We have removed the funding statement from the manuscript. 

"This research was based upon work supported in part by the Great Lakes Bioenergy Research Center, U.S. Department of Energy, Office of Science, Office of Biological and Environmental Research under Award Numbers DE-SC0018409. The research was conducted while the author was an employee of USDA-ARS and partly funded by congressionally allocated funds to the U.S. Dairy Forage Research Center."

Author response: Thank you for pointing this out. We wish to add the following to the Funding Statement: "The funders had no role in study design, data collection and analysis, decision to publish, or preparation of the manuscript." 

Author response: Thank you very much for this suggestion. We have submitted our data to Zenodo and it is currently being reviewed. Here is the URL to access the archive after it has been processed: 10.5281/zenodo.11169989. We have also added it to the Data availability statement (page 16).

5. Please review your reference list to ensure that it is complete and correct. If you have cited papers that have been retracted, please include the rationale for doing so in the manuscript text or remove these references and replace them with relevant current references. Any changes to the reference list should be mentioned in the rebuttal letter that accompanies your revised manuscript. If you need to cite a retracted article, indicate the article’s retracted status in the References list and also include a citation and full reference for the retraction notice.

Author response: Thank you for highlighting this. To the best of our knowledge, we have not cited any retracted papers in our manuscript. We have, however, added the following references in order to address some of the reviewer comments (these are also addressed directly in response to the specific comments): 

Johnson, Nancy Collins, Gail W. T. Wilson, Jacqueline A. Wilson, R. Michael Miller, and Matthew A. Bowker. “Mycorrhizal Phenotypes and the Law of the Minimum.” New Phytologist 205, no. 4 (2015): 1473–84. https://doi.org/10.1111/nph.13172.

Kuzyakov, Y., Friedel, J.K., Stahr, K., 2000. Review of mechanisms and quantification of priming effects. Soil Biol. Biochem. 32, 1485–1498. https://doi.org/10.1016/S0038-0717(00)00084-5

Riley, R.C., Cavagnaro, T.R., Brien, C., Smith, F.A., Smith, S.E., Berger, B., Garnett, T., Stonor, R., Schilling, R.K., Chen, Z.-H., Powell, J.R., 2019. Resource allocation to growth or luxury consumption drives mycorrhizal responses. Ecol. Lett. 22, 1757–1766. https://doi.org/10.1111/ele.13353

Stacey, G., Burris R., Evans H. Biological Nitrogen Fixation. New York, NY: Springer, 1992. https://link.springer.com/book/9780412024214.

Reviewer comments:

Review Comments to the Author

Reviewer #1: Dear authors,

this is an interesting work and can be published in this journal.

please present your Rescripts for diversity analyses (phyloseq,..) in supplementary file (GitHub link) and refer in the materials and methods.

Best regard

Author response: Thank you very much for your thoughtful feedback on this manuscript. We have added our R scripts to the same repository that we have made our data publicly available (10.5281/zenodo.11169989). We think that this will make accessing our data and scripts streamlined and efficient. We added a reference to the location of our R scripts to the manuscript as well (page 10, line 221). 

Reviewer #2: The MS by Stonoha-Arther et. al investigates microbial communities of switchgrass rhizosphere in response to nitrogen availibility in two different locations.The study identifies unique microbial taxa associated with nitrogen non-responsive switchgrass families and highlights the complex interactions between plant genotype, location, and soil microbial communities in switchgrass nitrogen response, the leads from which can be adapted for future breeding programs to reduce nitrogen use in switchgrass cultivation. However, the MS in its present forms needs to be revised suitably before accepted for publication. The major and minor comments attached as a word document

Author response: Thank you very much for your close reading of the manuscript. We have addressed the comments that came in the form of a Word document. 

Additional comments from reviewer 2:

Major Comments:

1. The introduction part needs major reaarngements, e.g. the role of microbes and Arbuscular mycorrhizal fungi in general may be concised and brought before throughly summerizring the role of these two in context of switchgrass. 

Author response: Thank you very much for this useful suggestion. We have substantially edited and rearranged the introduction to increase readability and relevancy. We took the specific suggestion about the paragraphs dedicated to the role of microbes and AMF and made those more concise (pages 3 and 4). 

2. The result section is an overlap and contains a lot of methodology, which needs to be suitably edited. Overall the the results are hard to follow throughout due to lack of proper presentation, it needs to elaborated for better readability.

Author response: Thank you very much for pointing this out. We have made significant edits throughout the results section in order to improve its usefulness and flow. We have taken out excess detail that belongs in the methods section (see pages 12 and 13 in the marked-up manuscript for an example). 

3. They identified some unique microbial taxa associated with nitrogen non-responders indicating genotype-specific interactions with rhizosphere microbiota. Further functional studies needs to be carried out to validate the mechanisms underlying these associations, as there is no correlation in nifH abundance between nitrogen-responsive and non-responsive groups raising query, if at all, there is any direct involvement of such microbial groups in mediating switchgrass response to nitrogen. Additional functional studies needs to be performed showing the roles of atleast a few specific taxa.

Author response: Thank you for this comment. We agree that further functional studies on some of the taxa outlined in this manuscript need to be followed up on. Future work will be focused on the possible functions that these taxa are performing for the switchgrass. For this study, we had the following objectives: to characterize the switchgrass rhizosphere microbiome with and without added nitrogen fertilizer; to identify microbial candidates that may contribute to the biomass accumulation of switchgrass genotypes that do not respond to nitrogen fertilizer; and to identify any microbes that may be preventing the non-responsive switchgrass genotypes from utilizing nitrogen fertilizer (page 4). While it outside the scope of this paper to include functional studies, we added the following to the discussion (page 14) to address the nifH qPCR results: “However, presence of DNA does not necessarily indicate a difference in expression of the gene. nifH gene expression could be higher in the non-responders and receive greater amounts of fixed nitrogen despite the lack of difference in nifH copy number or gene abundance.”

4. In one location, there was an increase in AMT1 gene abundance in response to nitrogen fertilization which is contradictory. Is this due to underlying complexity of plant-microbe interactions along with existing environmental conditions at that specific site? If the authors wish to keep this observation in manuscript, they need to provide logical explaination and hypothesis in support of this.

Author response: Thank you very much for pointing this out. We think these results are important to include in the paper and have added our hypothesis about what they mean to the discussion (page 15): “However, other studies have demonstrated that AMF can function as commensalists or parasites when plants are grown under nitrogen limiting conditions and the effect was dependent on plant, and, presumably, fungi genotypes [43,44].” These two cited papers were added during the revision. They have been added to the references: 

Johnson, Nancy Collins, Gail W. T. Wilson, Jacqueline A. Wilson, R. Michael Miller, and Matthew A. Bowker. “Mycorrhizal Phenotypes and the Law of the Minimum.” New Phytologist 205, no. 4 (2015): 1473–84. https://doi.org/10.1111/nph.13172.

Riley, R.C., Cavagnaro, T.R., Brien, C., Smith, F.A., Smith, S.E., Berger, B., Garnett, T., Stonor, R., Schilling, R.K., Chen, Z.-H., Powell, J.R., 2019. Resource allocation to growth or luxury consumption drives mycorrhizal responses. Ecol. Lett. 22, 1757–1766. https://doi.org/10.1111/ele.13353

Line 59: Expand AMF at its first use in the text 

Author response: Done! Thank you. New line is 52.

Line 79: Cite updated references 

Author response: Done! Thank you. New reference is:

Stacey, G., Burris R., Evans H. Biological Nitrogen Fixation. New York, NY: Springer, 1992. https://link.springer.com/book/9780412024214.

Line 89: AMT1 gene name should be italic 

Author response: Thank you for noticing this. This was actually taken out during our editing of the introduction. 

Line 105: Cite updated references 

Author response: Done! Thank you. The reference is:

Kuzyakov, Y., Friedel, J.K., Stahr, K., 2000. Review of mechanisms and quantification of priming effects. Soil Biol. Biochem. 32, 1485–1498. https://doi.org/10.1016/S0038-0717(00)00084-5

Line 110-11: Rephrase the sentence for better context 

Author response: Thank you for pointing this out. We have made the following changes (new line 89): “One reason that switchgrass is an attractive bioenergy crop is because it produces biomass on marginal soils. In order to further improve switchgrass for this purpose, it has been suggested that switchgrass should be bred to require less nitrogen while concomitantly selected for increased biomass in order to decrease the nitrogen removal that will inevitably increase if it is bred for biomass only [19].”

---

## [Decision Letter · Decision Letter 1]

14 Jun 2024

PONE-D-24-09855R1Rhizosphere microbial community structure in high-producing, low-input switchgrass familiesPLOS ONE

Dear Dr. Stonoha-Arther,

Thank you for submitting your manuscript to PLOS ONE. After careful consideration, we feel that it has merit but does not fully meet PLOS ONE’s publication criteria as it currently stands. Therefore, we invite you to submit a revised version of the manuscript that addresses the points raised during the review process.

We look forward to receiving your revised manuscript.

Kind regards,

Mayank Gururani

Academic Editor

PLOS ONE

Journal Requirements:

Reviewers' comments:

Reviewer's Responses to Questions

**Comments to the Author**

1. If the authors have adequately addressed your comments raised in a previous round of review and you feel that this manuscript is now acceptable for publication, you may indicate that here to bypass the “Comments to the Author” section, enter your conflict of interest statement in the “Confidential to Editor” section, and submit your "Accept" recommendation.

Reviewer #3: All comments have been addressed

Reviewer #4: (No Response)

2. Is the manuscript technically sound, and do the data support the conclusions?

Reviewer #3: Yes

Reviewer #4: Yes

3. Has the statistical analysis been performed appropriately and rigorously? 

Reviewer #3: Yes

Reviewer #4: Yes

4. Have the authors made all data underlying the findings in their manuscript fully available?

Reviewer #3: Yes

Reviewer #4: Yes

5. Is the manuscript presented in an intelligible fashion and written in standard English?

Reviewer #3: Yes

Reviewer #4: Yes

6. Review Comments to the Author

Reviewer #3: The manuscript titled "Rhizosphere Microbial Community Structure in High-Producing, Low-Input Switchgrass Families," authored by Christina Stonoha-Arther et al., provides valuable insights for the scientific community working on enhancing biomass production on marginal land for cost-effective bioenergy and forage production. The information presented in this manuscript will be of great interest to researchers in this field. The manuscript has been thoroughly revised based on previous Reviewer's comments, and it is now ready for acceptance.

Reviewer #4: Major Suggestions:

1.Reorganize the Introduction section: Condense the general information about the role of microbes and arbuscular mycorrhizal fungi (AMF), and focus more on summarizing their specific relevance in the context of switchgrass. This will improve the flow and relevance of the introduction.

2.Revise the Results section: There is an overlap between the Results and Methodology, which makes the results hard to follow. Separate the methodology details and present the results in a more structured and elaborated manner for better readability.

3.Perform additional functional studies: The paper identified unique microbial taxa associated with nitrogen non-responsive switchgrass families, suggesting genotype-specific interactions with the rhizosphere microbiota. However, the lack of correlation between nifH abundance and nitrogen responsiveness raises questions about the direct involvement of these microbial groups in mediating switchgrass response to nitrogen. Additional functional studies should be performed to validate the mechanisms underlying these associations and the roles of specific taxa.

4.Explain the contradictory increase in AMT1 gene abundance: The increase in AMT1 gene abundance in response to nitrogen fertilization at one location is contradictory to the expected trend. Provide a logical explanation and hypothesis to support this observation, considering the underlying complexity of plant-microbe interactions and the environmental conditions at that specific site.

Minor Suggestions:

1.Expand the abbreviation "AMF" at its first use in the text.

2.Cite the updated references for lines 79 and 105.

3.Rephrase the sentence on lines 110-111 for better context.

Overall, the paper presents an interesting study on the switchgrass rhizosphere microbiome in response to nitrogen availability, and the suggestions above should help improve the manuscript's readability, clarity, and the depth of the analysis.

7. PLOS authors have the option to publish the peer review history of their article (what does this mean?). If published, this will include your full peer review and any attached files.

Reviewer #3: **Yes: **Sajeesh Kappachery

Reviewer #4: No

---

## [Author Response · Author response to Decision Letter 1]

3 Jul 2024

Mayank Gururani and reviewers:

Thank you for your time and effort to improve this manuscript titled, “Rhizosphere microbial community structure in high-producing, low-input switchgrass families.” We sincerely appreciate the opportunity to edit this manuscript and we hope that we have addressed the comments to the satisfaction of the editors and reviewers. Here, we have included a point-by-point response to all of the comments that we have received for this second round of revisions. Changes are outlined here and in our marked-up copy of the manuscript. 

Reviewer comments:

Review Comments to the Author

Reviewer #3: The manuscript titled "Rhizosphere Microbial Community Structure in High-Producing, Low-Input Switchgrass Families," authored by Christina Stonoha-Arther et al., provides valuable insights for the scientific community working on enhancing biomass production on marginal land for cost-effective bioenergy and forage production. The information presented in this manuscript will be of great interest to researchers in this field. The manuscript has been thoroughly revised based on previous Reviewer's comments, and it is now ready for acceptance.

Author response: Thank you very much for your careful reading of the manuscript and your thoughts on this study. We sincerely appreciate the time that you took to review our paper and we hope that the scientific community finds this work useful. 

Reviewer #4: Major Suggestions:

1.Reorganize the Introduction section: Condense the general information about the role of microbes and arbuscular mycorrhizal fungi (AMF), and focus more on summarizing their specific relevance in the context of switchgrass. This will improve the flow and relevance of the introduction.

Author response: Thank you very much for this useful suggestion. We have substantially edited and rearranged the introduction to increase readability and relevancy. We took the specific suggestion about the paragraphs dedicated to the role of microbes and AMF and made those more concise (pages 3 and 4). 

2.Revise the Results section: There is an overlap between the Results and Methodology, which makes the results hard to follow. Separate the methodology details and present the results in a more structured and elaborated manner for better readability.

Author response: Thank you very much for pointing this out. We have made significant edits throughout the results section in order to improve its usefulness and flow. We have taken out excess detail that belongs in the methods section (see pages 12 and 13 in the marked-up manuscript for an example). 

3.Perform additional functional studies: The paper identified unique microbial taxa associated with nitrogen non-responsive switchgrass families, suggesting genotype-specific interactions with the rhizosphere microbiota. However, the lack of correlation between nifH abundance and nitrogen responsiveness raises questions about the direct involvement of these microbial groups in mediating switchgrass response to nitrogen. Additional functional studies should be performed to validate the mechanisms underlying these associations and the roles of specific taxa.

Author response: Thank you for this comment. We agree that further functional studies on some of the taxa outlined in this manuscript need to be followed up on. Future work will be focused on the possible functions that these taxa are performing for the switchgrass. For this study, we had the following objectives: to characterize the switchgrass rhizosphere microbiome with and without added nitrogen fertilizer; to identify microbial candidates that may contribute to the biomass accumulation of switchgrass genotypes that do not respond to nitrogen fertilizer; and to identify any microbes that may be preventing the non-responsive switchgrass genotypes from utilizing nitrogen fertilizer (page 4). While it outside the scope of this paper to include functional studies, we added the following to the discussion (page 14) to address the nifH qPCR results: “However, presence of DNA does not necessarily indicate a difference in expression of the gene. nifH gene expression could be higher in the non-responders and receive greater amounts of fixed nitrogen despite the lack of difference in nifH copy number or gene abundance.”

4.Explain the contradictory increase in AMT1 gene abundance: The increase in AMT1 gene abundance in response to nitrogen fertilization at one location is contradictory to the expected trend. Provide a logical explanation and hypothesis to support this observation, considering the underlying complexity of plant-microbe interactions and the environmental conditions at that specific site.

Author response: Thank you very much for pointing this out. We think these results are important to include in the paper and have added our hypothesis about what they mean to the discussion (page 15): “However, other studies have demonstrated that AMF can function as commensalists or parasites when plants are grown under nitrogen limiting conditions and the effect was dependent on plant, and, presumably, fungi genotypes [43,44].”

Minor Suggestions:

1.Expand the abbreviation "AMF" at its first use in the text.

Author response: Done! Thank you. New line is 52.

2.Cite the updated references for lines 79 and 105.

Author response: Done! Thank you. New reference for line 79 is:

Stacey, G., Burris R., Evans H. Biological Nitrogen Fixation. New York, NY: Springer, 1992. https://link.springer.com/book/9780412024214.

The reference for line 105 is:

Kuzyakov, Y., Friedel, J.K., Stahr, K., 2000. Review of mechanisms and quantification of priming effects. Soil Biol. Biochem. 32, 1485–1498. https://doi.org/10.1016/S0038-0717(00)00084-5

3.Rephrase the sentence on lines 110-111 for better context.

Author response: Thank you for pointing this out. We have made the following changes (new line 89): “One reason that switchgrass is an attractive bioenergy crop is because it produces biomass on marginal soils. In order to further improve switchgrass for this purpose, it has been suggested that switchgrass should be bred to require less nitrogen while concomitantly selected for increased biomass in order to decrease the nitrogen removal that will inevitably increase if it is bred for biomass only [19].”

---

## [Editor Report · Decision Letter 2]

30 Jul 2024

Rhizosphere microbial community structure in high-producing, low-input switchgrass families

PONE-D-24-09855R2

Dear Stonoha-Arther,

We’re pleased to inform you that your manuscript has been judged scientifically suitable for publication and will be formally accepted for publication once it meets all outstanding technical requirements.

Kind regards,

Mayank Gururani

Academic Editor

PLOS ONE
---

## [Editor Report · Acceptance letter]

13 Aug 2024

PONE-D-24-09855R2 

PLOS ONE

Dear Dr. Stonoha-Arther, 

I'm pleased to inform you that your manuscript has been deemed suitable for publication in PLOS ONE. Congratulations! Your manuscript is now being handed over to our production team.

Kind regards, 

on behalf of

Dr. Mayank Gururani 

Academic Editor

PLOS ONE